# Digital Transformation—Top Priority in Difficult Times: The Case Study of Romanian Micro-Enterprises and SMEs

Daniela Roxana Vuță *[ID], Eliza Nichifor [ID], Ioana Bianca Chițu [ID] and Gabriel Brătucu [ID]

Faculty of Economic Sciences and Business Administration, Transilvania University of Brașov, 500036 Brasov, Romania
* Correspondence: daniela.vuta@unitbv.ro

**Abstract:** Ever since the pandemic context accelerated technology adoption, the digital transformation of enterprises has become part of consumers' daily lexicon. The highly demanded necessity of companies to integrate digital solutions and provide services and goods in virtual spaces provokes both managers and academia to identify new perspectives. In this context, can digital transformation through marketing represent an option in difficult times? The authors aimed to reveal an answer to this question by researching the attitudes of 837 entrepreneurs involved in digital marketing activity before and during the pandemic, following their behaviour in the post-pandemic context. With quantitative and qualitative methods, unexpected results were unveiled. Financial losses or low performance in terms of profit or turnover do not negatively influence the attitude of the subjects towards digital marketing adoption. Moreover, the most unexpected result was the evolution of turnover from 2020 compared to 2019 as a determining factor for entrepreneurs' perceptions of their level of digital knowledge. However, the results are discussed in the context of digital transformation through the method of providing services and goods in the virtual space with digital techniques, enriching the literature with the Romanian micro, small and medium-sized enterprises evidence.

**Keywords:** digital transformation; micro-enterprises; SMEs; COVID-19; digital marketing





## 1. Introduction

Radical and disruptive changes have happened due to the COVID-19 pandemic. With its effects still reshaping the world, the pandemic made as clear as ever that companies have to prepare for operating in an increasingly uncertain and volatile environment to increase their sustainability [1,2]. To achieve this, they need to search for those enabling factors that can facilitate the process of decision-making, support a strong and indispensable asset portfolio, and strengthen its ordinary and dynamic capabilities. The study of Schoemaker [3] introduces ordinary capabilities as the main enablers for the process of innovation, while dynamic capabilities help the organization to monitor and analyse the external environment; thus, forecasting changes, reaching for opportunities in timely ways, and innovating the business model. According to Hachicha and Bekri [4], the ability to collect data, process them, and then convert them into usable information is an indispensable requirement to gain a competitive advantage and safeguard an organisation's survival in today's markets. Predictive analytics, artificial intelligence, machine learning, digital platforms, and big data are some of the tools that companies going through digital transformation (DT) are using to leverage their key resources, ensure business adaptability, and build resilience [2,5].

In a world where technology and the internet are evolving daily, both academic and business environments acknowledge digital transformation as a prerequisite "to improve business processes, products, services, and operations management" [2,6,7]. Some scholars see it as a process [8,9] that involves the following three stages: digitization, digitalization, and digital transformation [10]. While other researchers describe DT from a strategy perspective [6,11], most studies agree that implementing digital technologies involves

radical changes in business processes [12,13], organizational structure [2], marketing and customer service [14,15], or supply chain integration [16,17].

Moving forward with the economic implications of DT, special attention has been attributed to investigating the challenges of digitalization and digital technologies on social and environmental welfare [18]. Several studies highlighted the connection between social sustainability and DT [19], as well as the need for organisations to integrate sustainability strategies within their digital transformation roadmaps [20].

Although new digital technologies are available for all types of organizations, existing research suggests that digital transformation has multiple challenges and peculiarities for micro, small and medium-sized enterprises (SMEs) [21,22]. Micro-enterprises and SMEs (MSMEs) are seen as the driving force of a country's economy but are also more vulnerable compared to other businesses [23]. According to the United Nations, MSMEs represent nearly 95% of business companies worldwide, accounting for 60% of employment [24]. However, most of these businesses fail to reach their potential for digital development as they do not fully realize their innovative and creative capacity through technology [25]. This topic has become one of interest given that the digital age is constantly making its mark on both social and economic activities. While the concern for improving the relationship with customers and other stakeholders remains a strategic priority, how enterprises adapt to newer circumstances facilitates their success [26]. English translation. Extant literature presents the challenges faced by micro, small and medium-sized enterprises in the current context [7,27,28], namely, their need to carry out digitalization acquisitions within a business that do not show readiness for implementing a digitized model [29–31], and being less aware of its effects [13], through which not only the rules change but also how economic agents play on the market.

The concerns found in literature in terms of constraints and prerequisites for the digital transformation of SMEs refer to enterprise characteristics, bounded resources (financial and personnel) [32,33], privacy and security concerns [28], limited knowledge [34], customers, competitors, and technological barriers [12,35]. At the same time, the inter-dimensional dynamics between all subjects are described, shaping a solution for overcoming the short-comings of the markets in which micro-enterprises and SMEs operate [36]. Top management skills, knowledge, and support, as well as changes in business practices with the help of the internet, e-commerce, and social media, are among the factors that favour digital transformation in a sustainable way [23,37,38].

In this context, the authors wanted to find an answer to the following research question: "*Can digital transformation become a response for micro, small and medium enterprises to deal with a crisis?*". To find a plausible answer, the purpose of the study is to discover the behaviour of micro and SMEs entrepreneurs by performing a mix-method of research using qualitative and quantitative methods on the digital transformation topic during the hard times of the COVID-19 pandemic.

The paper is structured in six sections. After the introduction of the context and the literature review, the authors present the materials and methods used to conduct the study. Next, the results are shown in the fourth section and discussed in the extended following section. In the final section, the conclusions of the study highlight the scientific contribution, the managerial and academic implications, the limitations, and the future research directions.

## 2. Digital Transformation of SMEs through Digital Marketing

The context of the COVID-19 pandemic has seized the entire society and, therefore, a major domestic and global economic recession is to be expected [39]. In this context, the sustainability of SMEs is a focus area that gains more attention. Worldwide organizations, such as United Nations, acknowledge that MSMEs are facing some of the greatest economic difficulties due to the COVID-19 pandemic [24]. All these challenges constitute the general framework in which micro-enterprises and SMEs are carrying out their activities and which will mark them in the near future. Various scholars agree that there is still a general gap

in understanding the effects of COVID-19 on enterprises and their digital transformation journeys during a disruptive business environment [27,28,40–42].

Developing enterprise competitiveness and maintaining a long-term relationship with customers, both in the offline environment and in the online environment, are key factors influencing all the other strategic objectives [23]. With the rapid development of e-commerce and digital platforms [14], selling products relies more on digital marketing from the perspective of marketing orientation towards the customer and the market, in the context of harmonizing the company's objectives with understanding consumer needs and satisfaction [10,43]. Research shows that enterprises that do not focus their attention on the techniques and tools specific to digital marketing will not be able to capitalize on the diversity of methods by which to convey their message to target audiences, and the organization may face real challenges in the long run [15,44,45].

The use of digital marketing technologies drives the organization toward a digital transformation journey [46]. However, this is very difficult to achieve for entities that do not have enough financial resources or have limited know-how, given that they require digital acquisitions, training, hardware and software compatibility, maintenance, and a dedicated team [47]. The lack of digitalization generates a technology problem with data collection and analysis, which are extremely important in the adoption of digital marketing processes. This leads to decisions based on intuition [48], without support data, slowing down reactions over time and anticipation of trends in the industries in which enterprises operate [49], reactive actions by organizations, and a lack of precious information about consumer behaviour. Micro, small and medium-sized enterprises that do not have a presence in the digital environment need to invest in technology and adopt a complete digital transformation to remain sustainable [50–52].

## 3. Materials and Methods

In aiming to find an answer for the critical points, two types of methods were applied by applying a mixed research method. In the first stage, semi-structured in-depth interviews have been initiated to intentionally cover the pandemic and unveil the feelings of insecurity and the concerns of entrepreneurs on the uncertainties of the given context. The purpose of it was to understand the coordinates and the context of researching such a theme. Intending to discover the potential for the study, this was represented by the method of analysing the behaviour of the entrepreneurs in the preliminary phase. The outcomes depicted the questions and the research themes for the second study. In other words, the results of qualitative research generated quantitative research perspectives, by aiming to quantify the attitude of entrepreneurs regarding digital presence after dealing with the COVID-19 pandemic, as well as the main differences regarding the performances of micro, small and medium enterprises in the post-lockdown context. For a good understanding of the methodology of both studies, two subsections of materials and methods are presented. The first one is committed to qualitative research, while the second one is dedicated to quantitative research.

### 3.1. Semi-Structured in-Depth Interview

This qualitative method aimed to explore the perceptions of entrepreneurs regarding digital transformation in relation to certain concepts specific to online marketing, all in a full pandemic context. Thus, the attitude towards digital tool adoption, online advertising, and future intentions represented the foundation of the study. The interviews were conducted using the guide created based on 3 topics strongly related to the objectives of the study: (O1) Identifying the opinions of Romanian entrepreneurs regarding digital transformation as a response to hard times; (O2) determining the attitudes of entrepreneurs towards the adoption of digital marketing tools; (O3) positioning Romanian entrepreneurs concerning online advertising methods. Starting with these three objectives, the authors assumed that the entrepreneurs know the technologies that would facilitate the digital transformation process, that they consider the lack of knowledge in digital marketing as the main issue

for digital transformation, and that they continue their online promotion activity after the COVID-19 pandemic.

Firstly, the process of conducting the study involved recruiting entrepreneurs from Romania using online channels, such as social media (Facebook, Instagram, LinkedIn), or direct e-mail to participate in the interview. The only criterion for selection was that the entity would represent an existing micro-business or SME with online activity before the pandemic. After the recruiting process (which took two weeks), ten participants were selected for confirmation to take part in the discussion. The entrepreneurs were interviewed, individually, for 60 min, in video-conferencing mode. The investigation tool was designed in such a manner to obtain sufficient information to achieve the proposed objectives and to confirm or refute the assumptions presented above.

The research was designed to support the gathering of information as accurately as possible for the environment in which Romanian entrepreneurs operate, where digital presence is increasingly necessary. To this end, important aspects of the digital transformation of micro-enterprises and SMEs, the attitude towards the adoption of digital tools, and the positioning of online advertising in the pandemic context were addressed.

*3.2. Survey-Based Research*

Relying on the results of qualitative research, the authors decided to conduct quantitative research aimed at measuring the potential of micro-businesses and SMEs' digital transformations through digital marketing, while undergoing the effects of the COVID-19 pandemic. Related to this objective, the research questions for quantitative research were defined as:

1.  What is the attitude of entrepreneurs regarding the digital presence of the companies they run after the COVID-19 pandemic?
2.  What are the main differences in the performance of micro-enterprises and SMEs and the behaviour of entrepreneurs in the post-lockdown context?

Thus, the survey-based exploratory investigation was chosen, applying the non-random sampling method. The questions were represented by nominal, ordinal, and ratio scales, which demonstrate the validity and reliability of quantitative research design [53] The data collection process was carried out online, through the CAWI (Computer-Assisted Web Interviewing) method, by distributing the questionnaire on social media but also by volunteering to online groups and communities in the country. The researched population is represented by people over the age of 18, and all as entrepreneurs with an active legal form of micro-business and SMEs. The data collection process was completed by applying the filter of eligibility of respondents. In the end, 832 entrepreneurs were included in the study. After, Excel applications and SPSS (Statistical Package for the Social Sciences) software (IBM, Armonk, NY, USA), version 28.0.1.0 (15) were used for data processing. The assumptions used to ground the analysis refer to the attitude of the entrepreneurs during difficult times and the important aspects of digital transformation in relation to the financial performance recorded in the pandemic year (2020) compared with 2019. To perform such a comparative analysis, supplementary information was collected for each enterprise. The financial information is public and has been collected from the Risco provider for Romanian companies (Table 1).

**Table 1.** Financial data collected for comparative analysis.

| No. of Indicator | Indicator |
| --- | --- |
| 1 | Turnover (2019) |
| 2 | The change in 2019 turnover compared to the previous year (%) |
| 3 | Financial outcome (2019) |
| 4 | The change in 2019 financial outcome compared to the previous year (%) |
| 5 | Turnover (2020) |
| 6 | The change in 2020 turnover compared to 2019 (%) |
| 7 | Financial outcome (2020) |
| 8 | The change of 2020 financial outcome compared to 2019 (%) |

Source: Created by the authors based on the information provided by www.risco.ro.

Based on the collected data and strongly related to the focus of the study, three hypotheses were set up, considering that the financial performance of micro-businesses and SMEs influence certain intentions and perceptions of the entrepreneurs.

**Hypothesis 1 ($H_1$).** *There are no differences between entrepreneurs who want to intensify online activity with the Google My Business (hereinafter GMB) digital tool and those who do not, in terms of the evolution of 2020 turnover compared to 2019.*

**Hypothesis 2 ($H_2$).** *There is no link between the financial outcome of the pandemic year and the perceived importance of the company's activity in the online environment.*

**Hypothesis 3 ($H_3$).** *There is no link between the level of self-perceived knowledge and the evolution of turnover from 2020 to 2019.*

**4. Results**

*4.1. Qualitative Research Results*

The interview guide was structured according to each research theme and the results follow the same line to present the findings regarding digital transformation.

4.1.1. Entrepreneurs' Opinions on Digital Transformation through the Online Presence of the Business

Regarding this theme, discussions began with issues related to the digital transformation of business through online presence. Thus, the participants expressed personal opinions, examples from their own experiences, or testimonies.

Asked about the most suitable moment to begin a digital transformation of the business the entrepreneurs showed some concern for the digital transformation. Although the interviewees stated that *every company should exist on the Internet* or that *nowadays you do not exist if you do not show up online*, the participants stated that they do not know exactly what the right time would be to initiate a process of digital transformation. Rather, some of them have assumed that they care more about the digital presence when they *notice that sales are falling*, or *in the dead*. Furthermore, two of them had the courage to express that they do not know the relevance of the activity of constant promotion, because *the money invested never fully returns" or "customers do not come to buy online anyway"*. All these statements led the researchers to deepen the pain that entrepreneurs feel on this topic. In this regard, the interviewees were asked to provide more details about their efforts made in virtual space. In response, entrepreneurs stated that they feel able to communicate with ease on this topic, noting a certain potential to change their behaviour by promoting information on these concepts: "if I were to think better, now I understand how important it is to better document yourself before saying about a certain application whether it is useful or not". On the other hand, a subject expressed that he understands very well the need to transform his business, even wanting to improve communication with its customers online, showing great interest in a possible collaboration with a consultant in the field, *to explain more about what I can do.*

Regarding the technologies that would facilitate the digital transformation, the subjects generally listed digital promotion channels but did not specifically mention the applications they used at that moment. Only one subject expressed that the online presence he is trying to develop is conducted through a website, but he wants to improve it, not being very satisfied with the current version. Five interviewees mentioned Facebook and Instagram as the main platforms through which they can promote their business, but when asked if they use the two platforms at the time, three of them said they only have a personal account for each platform. Only two subjects discussed the trend of using audio-video media to enhance the online presence of the business.

Next, the participants were asked about the advantages and disadvantages of increasing the online presence of the business. On this topic, participants showed that they agree with the online environment: "yes, of course, you must exist there", but they quickly moved on, considering this topic of discussion *very annoying*. On the other hand, one participant said that he would like to learn everything, if possible, about how to work more in the online environment, but a major disadvantage is the *lack of courses in Romanian* and *the use of so many applications considered necessary for my company, that it is simply difficult for me to choose what is right for the strategy. This may have stopped me from wanting to discover more*. Another disadvantage expressed by a subject on this topic refers to the role of the entrepreneur *to make the company work, not having time to worry much about promoting the company*. The same disadvantage was expressed by another subject but, in different terms: *I don't have time to deal with all these things, and a person hired just for that, it would be too much. He would have nothing to do eight hours a day*. However, four of the ten subjects strongly stated that *finding new customers*, *brand awareness*, and *increased visibility* are among the competitive advantages obtained by a business if they expose their company online. Regarding the other entrepreneurs, it can be said that they did not necessarily feel comfortable being asked about the advantages or disadvantages in this field, not assuming a formulated opinion.

Through hypothetical discussion, the entrepreneurs were encouraged to imagine what the digital presence of their business would ideally look like, observing a change in perception of the digital presence. Almost all of them expressed that, ideally, they would have an intense digital presence, *with constant activity on social media*, *with frequent posts* and *a well-developed strategy*. Furthermore, another perspective from which the subjects looked at the issue was the budget allocated to *paid posts*, *online campaigns*, but also the profitability of the expenses allocated to increase the digital presence: "for me, the ideal digital presence would bring money back as soon as possible. I am tired of the specialists who say that recovery of money can't happen shortly after the beginning of the campaigns ", according to one of the participants. Another subject said that "from my point of view, a good online presence generates customer trust. That's what I would like if I could stick to one strategy".

### 4.1.2. Entrepreneurs' Attitudes toward the Adoption of Digital Marketing Tools in the Post-Lockdown Context

Asked how prepared they are to use digital marketing tools to promote their business, the results show that in at least half of the cases, entrepreneurs stated that they do not feel very prepared to use certain digital marketing tools to promote their business. Some of them said that this aspect depends on the type of tool, identifying the ones used as easy to use: *It depends on which tool it is. If we talk about Facebook and Instagram, I have no problems. If it is about something more advanced, I would need time to learn*. Another subject said that when it comes to what he is already using, he feels as if *I know that I would have to work until I master them*. Three of the interviewees confirmed that they need help to feel ready to promote the business (*I admit I need help*; "*I want to learn more, but in my way*"; *I confirm that I need help. Although I like to believe that I keep up, I am not dealing with them*).

Another question was addressed to identify the main problems that entrepreneurs face when they want to deploy a new way to promote the business in the virtual space. The level of knowledge needed to implement ways to increase the online presence is one of the main reasons why entrepreneurs do not perform in this regard. One statement drew attention to

the role of permanent presence in the online environment (*Perseverance. For some reason, we stop. We stop posting, communicating news to our audience, and this is not beneficial to the business at all*), and another specifies the desire to know better the responsibilities of specialists in the field (*Communicating with specialists in the field. Are we looking for IT specialists or marketers to help us?*). The respondent's answer is also part of the category of knowledge level (*I think we don't know what to change. We don't know what doesn't work or what could work better*), which refers to the immediate need to train entrepreneurs on the adoption of digital marketing tools.

Another notable result is that no entrepreneur responded by mentioning at least one digital tool for organic promotion. Compared to paid advertising, the interviewees expressed various feelings, but only one subject mentioned running campaigns. Moreover, *we are satisfied, but an advanced analysis of the online public would help us even more to streamline the allocated budget*. Another participant also stated about the budget, and two more consider it inappropriate for adopting this method (*I think it works when you know exactly what to promote. In the current context, in our case it does not seem relevant to do paid advertising; I don't know exactly what my position is on paid advertising, but I consider it effective for larger companies*.). Instead, an interesting opinion was expressed by another participant (*it is effective when you already know who to show your ads to*). Only one subject mentioned an unpleasant experience with a specialist in the field, which led him to stop online advertising campaigns.

Next, the participants were invited to discuss what is the trend of specialized digital marketing services in the context of the COVID-19 pandemic. In this regard, all the entrepreneurs believed that in the future, specialized digital marketing services *will be growing, as there is a growing need for people to help companies grow*. A participant mentioned the quality of services, saying that *on the other hand, it will be easy to choose in terms of the quality offered. I noticed the ease with which some people start to offer such services and although there will be more and more, few will offer a high quality of the services*.

One of the participants intended to intensify the promotion activity, *no matter what happens in the future*. The pandemic context, however, was the reason for the other interviewees to doubt the action of allocating resources for online marketing activity for reasons of insecurity and fear of running out of financial resources. *Although we are aware of the importance of promoting online during this period, it may become difficult to maintain a well-promoted image if sales decrease, If my service is not tradable online, I will not be able to continue to promote it during this period except through unpaid methods*. Unfortunately, following the discussions, the marketing activity was not on the list of priorities that entrepreneurs would pay in the pandemic context, considering that it is much more important to survive in the market than to intensify online promotion. In eight out of ten cases, it was observed that the "fault" for such a decision would be the "current context". Only for two of the entrepreneurs was it important to allocate resources in the direction of digital development, stating that *No matter what happens after this lockdown, I will continue to promote myself online*, respectively, *we hope to be able to maintain the online promotion activity. Given the field in which we operate, we are optimistic*.

Uncertainty and fear of the unknown in the next period of pandemic context were two recurring aspects in the individual discussions with entrepreneurs (*We do not know how long customers will be relaxed to buy. I feel that at some point they will start not spending money. For the moment, I do not feel a crisis, but it will happen*).

*4.2. Quantitative Research Results*

The following subchapter contains the results of the research, briefly presented through the preliminary analysis, but also the testing of hypotheses. The outcomes are presented to facilitate data transformation into useful information for decision-makers, introducing frequencies, graphical tables, or relevant indicators. The analysis was performed using Excel and SPSS and is based on 35 variables generated according to the items of the questionnaire, excluding the first filter question.

The results of the quantitative research are presented according to the assumed hypothesis, as follows.

To test the first hypothesis, the Mann–Whitney U test was applied. By running the test with SPSS, the results presented in Table 2 were obtained.

**Table 2.** Mann–Whitney U test results.

| Test Statistics [a] | |
|---|---|
| Mann–Whitney U | 78,390.00 |
| Z | −0.144 |
| Asymp. Sig. | 0.885 |

[a] Grouping variable: answers for intensifying online activity. Source: created by the authors based on the SPSS output.

Given the value of the significance level Asymp. Sig. (0.885), which is higher than the level of significance considered (0.05), it can be stated the acceptance of the null hypothesis, $H_0$, according to which there are no differences between entrepreneurs who want to intensify online activity with the GMB digital tool and those who do not want to, in terms of the evolution of 2020 turnover compared to 2019.

For the second hypothesis, the authors considered enterprises' financial performances in relation to entrepreneurs' perceptions of the importance considered for online activity and referred to the link between the financial result in 2020 and the perceived importance of a company's activity in the online environment.

For $H_2$, the Kolmogorov–Smirnov statistical test was selected for bivariate analysis. Therefore, the differences between the observed and expected distributions, which cumulatively increased, were tested for 832 of 837 entrepreneurs that answered the question regarding the importance of online presence (Table 3).

**Table 3.** Contingency table regarding the financial performance of the enterprise in relation to the online activity importance perceived by entrepreneurs.

| Contingency Table | | | | | | |
|---|---|---|---|---|---|---|
| | | | Financial Outcome | | | Total |
| | Importance | | 0 | Profit | Loss | |
| How important do you think the online presence of the business is? | 2 | Responses | 15 | 12 | 33 | 60 |
| | | % In the category financial result 2020 | 8.8% | 6.0% | 7.2% | 7.2% |
| | 3 | Responses | 12 | 26 | 51 | 89 |
| | | % In the category financial result 2020 | 7.1% | 12.9% | 11.1% | 10.7% |
| | 4 | Responses | 38 | 45 | 108 | 191 |
| | | % In the category financial result 2020 | 22.4% | 22.4% | 23.4% | 23.0% |
| | 5 | Responses | 105 | 118 | 269 | 492 |
| | | % In the category financial result 2020 | 61.8% | 58.7% | 58.4% | 59.1% |
| Total | | Responses | 170 | 201 | 461 | 832 |
| | | % In the category financial result 2020 | 100% | 100% | 100% | 100% |

Source: Created by authors based on the SPSS output.

As can be seen in the table, there are differences between the two groups (profit and loss) in terms of frequency distributions. To determine the statistical significance, the Kolmogorov–Smirnov test was applied to test the maximum difference between the cumulative frequencies for the companies that recorded profit (F1) and for those that recorded loss (F2) is zero.

Thus, the calculation of differences was used (Table 4).

**Table 4.** Differences between the cumulative frequencies of the companies that registered profit and loss.

| | Relative Frequencies | | Cumulative Relative Frequencies | | Difference |
|---|---|---|---|---|---|
| 2 | 6.0% | 7.2% | 6.0% | 7.2% | −1.20% |
| 3 | 12.9% | 11.1% | 18.9% | 18.30% | **0.60%** |
| 4 | 22.4% | 23.4% | 41–30% | 41.70% | −0.40% |
| 5 | 61.8% | 58.7% | 103.10% | 100.40% | −2.70% |
| Total | 100.00% | 100.00% | - | - | |

Source: Created by authors based on the SPSS output.

Given that the value of $D_{calculated} = 0.60\%$ is less than the value of $D\alpha = 11.49\%$, the null hypothesis is accepted, according to which the maximum difference between the cumulated frequencies for the companies that registered profit (F1) and for those that registered loss (F2) is zero. The same decision can be made based on the results generated with the SPSS program (Table 5).

**Table 5.** Differences between the cumulative frequencies regarding the companies that registered profit and loss.

| | | "The Perceived Importance of the Company's Activity in the Online Environment" |
|---|---|---|
| | Absolute | 0.012 |
| Extreme differences | Positive | 0.012 |
| | Negative | −0.007 |
| Kolmogorov–Smirnov Z | | 0.141 |
| Asymp. Sig. (2-tailed) | | 1.000 |

Source: Created by authors based on the SPSS output.

The third hypothesis was tested to verify the link between the level of self-perceived knowledge and the evolution of turnover from 2020 to 2019:

By calculating the frequencies, some differences between groups were observed, which means the possibility of a connection. Therefore, the chi-square test was applied for bivariate analysis (Table 6).

**Table 6.** Chi-square to test the link between the self-perceived level of knowledge and the evolution of turnover from 2020 to 2019.

| 2020 Turnover Evolution Compared to 2019 Turnover Evolution | |
|---|---|
| Chi-square | 13,293.822 |
| df | 142 |
| Asymp. Sig. | 0.000 |

Source: Created by authors based on the SPSS output.

The results obtained can guarantee, with a probability of 95%, that there is a link between the level of self-perceived knowledge and the evolution of turnover in 2020 compared to 2019. The value of $\chi^2 = 13,293,822$ less than $\chi_{20,\ 0.05,\ 142}^2 = 170.80$ (calculated with the CHIINV function in Excel), together with the Asymptomatic Significance level (2-Sided < 0.05) determines the acceptance of the $H_1$ hypothesis.

Another interesting result was the increase in the proportion in cases of using the tool that ensures the minimum online presence. Although Google My Business is still used in a proportion of less than 30% (Figure 1a), at the beginning of the pandemic it recorded a much lower percentage of use, 11.90% (Figure 1b).

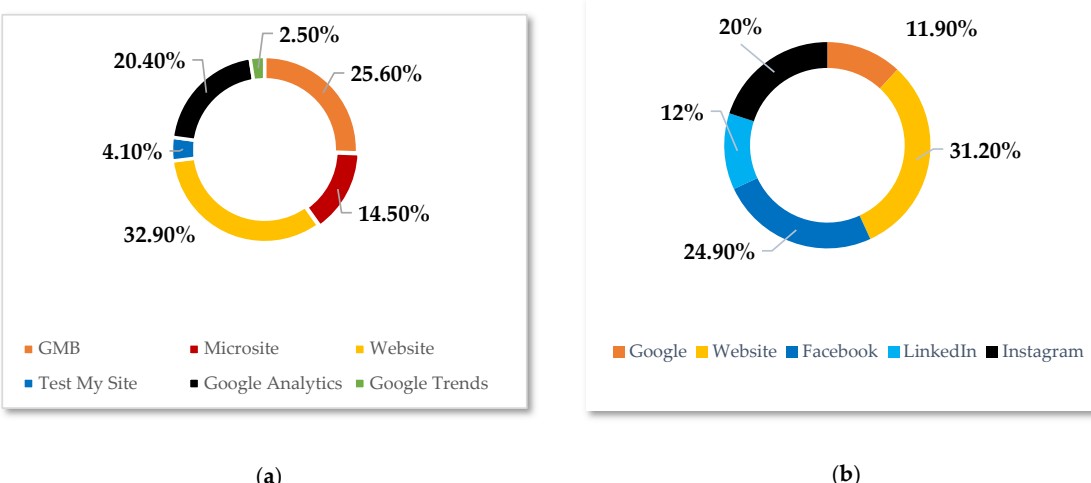

(**a**)                                                              (**b**)

**Figure 1.** Digital tools use comparison: (**a**) (pandemic period); (**b**) (immediately after the lockdown).

## 5. Discussion

This study aimed to explore the sustainable potential throughout the perceptions of entrepreneurs, as representatives of Romanian micro-enterprises and SMEs, regarding digital transformation, in conjunction with certain concepts specific to digital marketing, all in a full pandemic context. The literature review revealed that these links are not sufficiently investigated [27,28,40–42], but that they may be important in understanding attitudes toward digital tool adoption, online advertising, as well as future intentions and decision-making patterns for digital technologies implementation.

For a better understanding of the research, the authors created Figure 2.

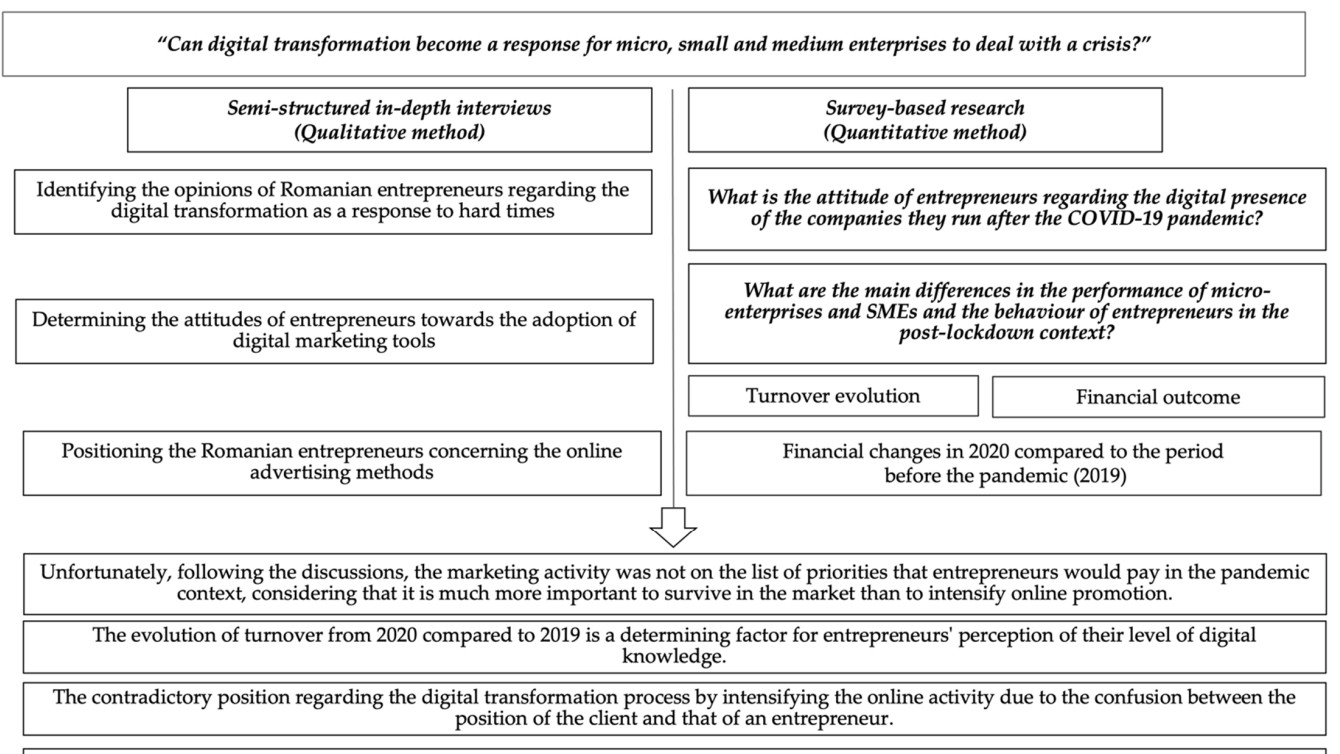

**Figure 2.** Research design.

Based on the results, the authors can state that, according to [49], specific patterns of digital technology adoption are related to the cognitive responses of entrepreneurs to digital transformation and their attitude toward opportunities. In addition to the mental barrier that the entrepreneurs have exposed concerning development opportunities of their business, they do not perceive the usefulness of advisory services in the field, considering it difficult to manage the time allocated by a person exclusively for digital development.

Moreover, the paper supplements the literature on the exploration of the barriers to the digital transformation of MSMEs [12,28,32–36] by investigating perceptions of fear or risk that entrepreneurs have regarding the initiation of online digital initiatives. As results indicated, one of the most important barriers is the perceived inability to predict the sustainability of digital initiatives due to the unpredictable negative influence of various resources (financial, time, human resources, current pandemic context). Having as a reference the period that changed the field of digital marketing (COVID-19 pandemic) [54], this study reveals that the evolution of the pandemic year turnover does not influence the intention to intensify online activity through the online presence of digital tools. This result is highly remarkable considering that previous studies have found that digital marketing initiatives have a significant effect on financial performance [55,56]. In the same register, the financial result for 2020 does not determine any change of perception among Romanian entrepreneurs regarding the importance of a company's activity in the virtual space. On the other hand, the evolution of turnover from 2020 compared to 2019 is a determining factor for entrepreneurs' perceptions of their level of digital knowledge.

The most unexpected result was the continuation of online promotion activities during the COVID-19 pandemic. Although entrepreneurs have shown an awareness of the significant role of online presence in the current context, most entrepreneurs have stated that they have diminished or given up the promotion activity due to financial uncertainty. The study unveiled the contradictory position regarding the digital transformation process by intensifying online activity due to the confusion between the position of the client and that of an entrepreneur.

## 6. Conclusions

The scientific contribution presented by the two types of marketing research methods places digital transformation through digital marketing as a top priority for Romanian micro-enterprises and SMEs, to gain sustainable advantages during hard times. Given the results, the study grounds a solid foundation for conducting other research in the field, bringing value primarily to academia, as new research opportunities can also be identified in the fields of entrepreneurship and digital transformation through digital marketing.

To the same extent, managerial implications are notable because specialists in the field, companies, or business consultants have the opportunity to understand the challenges of micro, small, and medium-sized enterprises, therefore developing strategies with concrete actions to mitigate them.

From the perspective of qualitative research, the impossibility of extrapolating the results to the researched population determines one of the first limits. Even so, considering the particularity of this type of marketing research, it can be said that the results obtained gave rise to general hypotheses for a subsequent quantitative study, presented in the second part of the paper. Another limitation refers to the high subjectivity generated by asking some questions considered relevant by the researcher, there being the possibility of omitting some undiscovered aspects due to the structure of the interview guide. Considering that the quantitative marketing research is based on a non-random sampling method, the lack of possibility to extrapolate the results obtained to the entire studied population is to be considered.

A further research direction may be a follow-up of the study to ensure representativeness in the target population. Subsequently, it can be improved by introducing several other questions in the questionnaire, which aims to analyse the behaviour of entrepreneurs toward digital transformation, following their target audience's reactions to the online

digital marketing initiatives. Finally, this marks the start of new extensive research aimed at studying what leverages are needed for an enterprise to start organizational capability building and induce digital marketing strategic changes through digital transformation.

**Author Contributions:** Conceptualization, D.R.V., E.N., I.B.C. and G.B.; methodology, E.N. and G.B.; literature review, I.B.C. and D.R.V.; analysis and writing the results, E.N; discussion and conclusions, E.N., I.B.C., D.R.V. and G.B.; writing—original draft preparation, E.N. and I.B.C.; writing—review and editing, I.B.C., D.R.V. and G.B.; supervision, G.B.; project administration, G.B.; funding acquisition, G.B. All authors have read and agreed to the published version of the manuscript.

**Funding:** The APC was funded by Transilvania University of Brasov.

**Institutional Review Board Statement:** Ethical review and approval were waived for this study, due to the absence of sensitive data and to the processing of data by ensuring confidentiality and anonymization of the personal information for all the subjects involved in the study.

**Informed Consent Statement:** Informed consent was obtained from all the participants in this study.

**Data Availability Statement:** Available on request.

**Conflicts of Interest:** The authors declare no conflict of interest.

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
