# Peer review of "Digital Transformation—Top Priority in Difficult Times: The Case Study of Romanian Micro-Enterprises and SMEs"

_sustainability, doi:10.3390/su141710741_

Round 1

Reviewer 1 Report

Dear Authors;

The research touches on an excellent topic; however, the method has some problems. What is your purpose? Hypothesis testing? Or an exploratory study? You must decide. If hypothesis testing is to be done, your methods are insufficient. These problems need to be resolved.

There are different approaches to how the scale should be developed in the research process. If the purpose of this study is to develop a scale, the method you used is not valid. Developing a scale in this way questions the validity and reliability of the study.

How were the questions you used in the survey method developed? Are these scales available in the literature? The validity and reliability of the scales were not specified in the research.

If you intend to develop a scale, you will come across different studies when you type "scale development process" and search on google images. (for example, Construct Measurement and Validation Procedures in MIS and Behavioral Research: Integrating New and Existing Techniques - Figure 1, Scott B. MacKenzie, Philip M. Podsakoff and Nathan P. Podsakoff). You should develop scale with this kind of flow process.

Another alternative is to divide your work into two. You can develop the qualitative part as an article and scale development as a separate study.

Good luck

Reviewer 2 Report

Review of “Digital Transformation - Top Priority in Difficult Times. The Case Study of Romanian Micro-Enterprises and SMEs “

The paper concerns the Digital Transformation of SMEs during and after the Covid-19 pandemic, with a particular focus on the Romanian business context. Specifically, the manuscript tries to answer to the following RQ: “Can digital transformation become a response for micro, small and medium enterprises to deal with a crisis?”.

I think this issue is really interesting and might attract much more interest in the near future. The paper is well-organized and quite clear. The research appears sound thanks to 832 entrepreneurs involved in the study. However, with this review, I can underline some minor issues that I suggest the authors to solve before the publication.

Minor points revealed by my review are listed below:

-        Although the article is understandable, its written form may be improved, e.g. sentence form, the excessive use of question marks and brackets (the use of italics may solve the problem?), and some English mistakes.

-        I would change the title of the first section “Digital transformation implications for SMEs” to “Introduction” since it introduces the problem and the aim of the paper with the linked RQ as an introduction typically does.

-        I suggest to add some more references related to digital transformation and social sustainability concepts. This issue is in fact strongly related to what this research explores. I would suggest the following references: - Grybauskas, A., Stefanini, A., & Ghobakhloo, M. (2022). Social sustainability in the age of digitalization: A systematic literature review on the social implications of industry 4.0. Technology in Society, 101997. - Linkov, I., Trump, B. D., Poinsatte-Jones, K., & Florin, M. V. (2018). Governance strategies for a sustainable digital world. Sustainability, 10(2), 440.

-        It is unclear the number of entrepreneurs involved. I suppose it is 832 as reported in Table 3. However, in the abstract it is reported the number of 837. 

Reviewer 3 Report

Paper 1839615 review to Sustainability – Digital Transformation - Top Priority in Difficult Times. The Case Study of Romanian Micro-Enterprises and SMEs

All issues raised in this review can be considered to be minor reviews.

General considerations

The subject of digital transformation is very relevant and current, and this article is an asset to highlight companies that are committed to survive in the business markets in which they operate. The article is very well structured, the contents explained with a good level of depth and well-articulated with each other. The data collection method and the analysis of the results obtained are presented in a perceptible and complete way, about the type of questions that integrate the interviews made to the entrepreneurs. All issues raised in this review can be considered to be minor reviews. However, a better adaptation of some parts of the text to the issue of sustainability (focus of the journal) would be something to improve on the authors' part.

1.   Structure

The structure of the article is well elaborated, but some flaws detected in the numbering of the sections and subchapters presented, namely in lines:

·         170 - The numbering of this subsection should be 3.2;

·         458 - The numbering of this section should be 6.

2.   Title, Abstract and Keywords

·         The title is appealing to readers.

·         The abstract is well constructed. The main research question, the objectives and the development of the theme are clearly pointed out.

·         The keywords are adequate.

3.      Figures and tables

The figures and tables are all well numbered, and have good visual quality.

4.      Grammar, spelling and syntax issues

The whole article it's well written in terms of grammar and spelling. But there were identified some aspects that should be improved/corrected, namely:

·         Whenever a term with its acronym is used, the first letters of each word must be capitalized;

·         In line 45 - the word "digitalization" appears twice...and it must be 3 different steps;

·         In line 147 - 5 discussion topics are mentioned, but only 3 are displayed. For readers who read the article sequentially, there seems to be a confusion between the "discussion topics" and the "objectives of the study";

·         In line 183 - the acronym CAWI must be accompanied by its full meaning, as this is the first time it is mentioned in the article;

·         In line 189 - the acronym SPSS must be accompanied by its full meaning, as this is the first time it is mentioned in the article;

·         In line 202 - the acronym GMB must be accompanied by its full meaning, as this is the first time it is mentioned in the article;

·         In lines 377 and 379, the numbering of hypotheses is the same as that of the hypotheses presented earlier in the article and with different designations. This is confusing for readers, so the numbering of all hypotheses could be sequential;

·         Comparing lines 357 and 377, the meaning of the Ho hypothesis is unclear. Therefore, the idea of sequential numbering is reinforced.

5.      Semantic and technical issues

The entire article is very well explained. The issues are explained very clearly and the concepts and ideas are very well articulated between themselves. The data collection method is explained clearly and objectively. The qualitative and quantitative analyzes are presented in a perceptible way, and Figure 2 was very well designed, summarizing both approaches to the results obtained. However, authors should adapt some sentences at the beginning and/or at the end of the sections of the article, for a better adaptation of the topic addressed in the article to the issue of sustainability - focus of the journal.

6.      References

The list of references is well prepared, the number of references is appropriate to the depth of the theme's approach in the article. The references are strong in the scope of this investigation. But throughout the text, the authors indicate the references in two ways simultaneously: with year-names and with numbers; and if the authors must reference in the first way, then for 4 or more authors they must use the term "et al." in italics.

Round 2

Reviewer 1 Report

In my opinion, the method used in the study is not sufficient for a scientific journal in terms of validity and reliability. I saw that the corrections I had mentioned in the first round were not made. 

This article needs serious effort to be corrected. The results should be supported by data collected from different sources whose validity and reliability have been proven statistically.

Round 3

Reviewer 1 Report

Dear Authors;

Most serious journals today no longer accept even articles with one-time survey methods, which are more reliable than qualitative methods. They request the results of the second survey to verifying analyze results. Your topic is good, but you must put more effort into verifying your results.